# Plasticity Comparison of Two Stem Cell Sources with Different Hox Gene Expression Profiles in Response to Cobalt Chloride Treatment during Chondrogenic Differentiation

**DOI:** 10.3390/biology13080560

**Published:** 2024-07-24

**Authors:** Sahar Khajeh, Vahid Razban, Yasaman Naeimzadeh, Elham Nadimi, Reza Asadi-Golshan, Zahra Heidari, Tahereh Talaei-Khozani, Farzaneh Dehghani, Zohreh Mostafavi-Pour, Masoud Shirali

**Affiliations:** 1Bone and Joint Diseases Research Center, Shiraz University of Medical Sciences, Shiraz 71348-14336, Iran; saharkh7@gmail.com; 2Department of Molecular Medicine, School of Advanced Medical Sciences and Technologies, Shiraz University of Medical Sciences, Shiraz 71348-14336, Iran; razban_vahid@yahoo.com (V.R.); yasamannaeimzadeh@gmail.com (Y.N.); nadimi.sums@yahoo.com (E.N.); zh.heidari66@yahoo.com (Z.H.); 3Stem Cell Technology Research Center, Shiraz University of Medical Sciences, Shiraz 71348-14336, Iran; 4Department of Anatomy, School of Medicine, Tehran University of Medical Sciences, Tehran 14166-34793, Iran; asadigr@ymail.com; 5Tissue Engineering Laboratory, Department of Anatomy, School of Medicine, Shiraz University of Medical Sciences, Shiraz 71348-14336, Iran; talaeitkh@gmail.com; 6Department of Anatomy, School of Medicine, Shiraz University of Medical Sciences, Shiraz 71348-14336, Iran; dehghanf@sums.ac.ir; 7Histomorphometry and Stereology Research Centre, Shiraz University of Medical Sciences, Shiraz 71348-14336, Iran; 8Department of Biochemistry, School of Medicine, Shiraz University of Medical Sciences, Shiraz 71348-14336, Iran; zmostafavipour88@yahoo.co.uk; 9Maternal-Fetal Medicine Research Center, Shiraz University of Medical Sciences, Shiraz 71348-14336, Iran; 10School of Biological Sciences, Queen’s University Belfast, Belfast BT9 5AJ, UK; 11Agri-Food and Biosciences Institute, Hillsborough BT26 6DR, UK

**Keywords:** chondrogenic differentiation, stem cell plasticity, *Hox* genes, dental pulp stem cell, bone marrow mesenchymal stromal cell, cobalt chloride

## Abstract

**Simple Summary:**

When joints are harmed due to injury, inheritance, or aging, it is difficult for the body to heal damaged cartilage. This is a major cause of pain and disability worldwide. With increasing life expectancy, millions of adults are affected, and enormous costs are imposed on health care systems. Scientists are attempting to use stem cells to create new cartilage-like tissue as a potential solution. However, the most suitable stem cell type for cartilage repair has not been defined. This study compared the ability to generate cartilage-like tissue by using two stem cell types that were derived from human bone marrow and dental pulp. To produce cartilage, the tissue’s environment should be simulated for stem cells in a laboratory. Therefore, the low-oxygen tension of joint cartilage was simulated using a simple and inexpensive chemical, cobalt chloride. The results showed that dental pulp stem cells could produce higher quality tissue that closely resembled human joint cartilage in terms of its structure and organization that could form the primary base of normal function. While further research is necessary, these initial findings represent a promising step toward using stem cells more beneficially to enhance a patient’s quality of life.

**Abstract:**

The limited self-repair capacity of articular cartilage is a challenge for healing injuries. While mesenchymal stem/stromal cells (MSCs) are a promising approach for tissue regeneration, the criteria for selecting a suitable cell source remain undefined. To propose a molecular criterion, dental pulp stem cells (DPSCs) with a *Hox*-negative expression pattern and bone marrow mesenchymal stromal cells (BMSCs), which actively express *Hox* genes, were differentiated towards chondrocytes in 3D pellets, employing a two-step protocol. The MSCs’ response to preconditioning by cobalt chloride (CoCl_2_), a hypoxia-mimicking agent, was explored in an assessment of the chondrogenic differentiation’s efficiency using morphological, histochemical, immunohistochemical, and biochemical experiments. The preconditioned DPSC pellets exhibited significantly elevated levels of collagen II and glycosaminoglycans (GAGs) and reduced levels of the hypertrophic marker collagen X. No significant effect on GAGs production was observed in the preconditioned BMSC pellets, but collagen II and collagen X levels were elevated. While preconditioning did not modify the ALP specific activity in either cell type, it was notably lower in the DPSCs differentiated pellets compared to their BMSCs counterparts. These results could be interpreted as demonstrating the higher plasticity of DPSCs compared to BMSCs, suggesting the contribution of their unique molecular characteristics, including their negative *Hox* expression pattern, to promote a chondrogenic differentiation potential. Consequently, DPSCs could be considered compelling candidates for future cartilage cell therapy.

## 1. Introduction

Articular cartilage defects resulting from trauma, inheritance, or age-related degeneration are major causes of global disability. Due to increasing life expectancy, these disabilities impose substantial costs on healthcare systems [1]. Despite its durability, articular cartilage demonstrates a remarkable inability to heal from injury and the deterioration caused by common trauma and diseases such as osteoarthritis (OA) due to its limited innate self-repair capacity. In the absence of effective pharmacological approaches, novel cell-based therapies need to be developed for cartilage regeneration with appropriate biochemical and structural features [2,3].

Multiple in vitro and in vivo studies have indicated the great promise of MSCs in regenerating injured tissue [4,5,6,7]. While several different potential sources of MSCs have been proposed for cartilage repair, it remains unclear which source is superior for this purpose.

BMSCs, the gold standard of MSCs, represent the most thoroughly characterized population of MSCs [8]. However, DPSCs have been gaining favor due to their accessibility and higher proliferative capacity compared to other sources [9]. Moreover, the ease of isolating human DPSCs with minimal pain and morbidity makes these cells a more valuable source than the classic BMSCs, rendering them extremely promising for establishing stem-cell-based therapies [10,11,12].

The chondrogenic differentiation of MSCs is influenced by various factors, including growth factors, cytokines, 3D scaffolds, and oxygen tension. However, there is currently no consensus on the optimal culture conditions required to drive MSC differentiation towards a stable chondrocyte phenotype [2,13,14,15].

The in vitro culture and differentiation of MSCs are typically performed under normoxic conditions, with a significantly higher oxygen tension compared to an in vivo level. This has failed to recapitulate the embryonic differentiation pathways of articular chondrocytes. Hypoxic conditions or low oxygen tension could imitate the physiological avascular microenvironment of articular cartilage [16,17,18]. Moreover, transplantation in a hypoxic tissue environment may necessitate the ex vivo hypoxia preconditioning of stem cells to facilitate the adaptation of the cells cultured under a high oxygen tension [18].

Hypoxia exerts its beneficial effects by enhancing matrix production, mitigating hypertrophy, and maintaining a chondrocyte phenotype through HIF-1α, the master regulator of the cellular response to low oxygen tension [17,18]. Additionally, hypoxia and hypoxia-mimicking agents, such as CoCl_2_, have been shown to positively influence the maintenance of stemness and the differentiation capacities of stem cells [2,7,19]. Similar to low oxygen tension, CoCl_2_ mimics hypoxia in vitro by preventing the degradation of the HIF-1α protein [18]. Notably, CoCl_2_ is preferred over low oxygen tension due to its low cost and ease of use, providing higher oxygen tension stability with no requirement for specialized incubators or chambers [18].

Cell behaviors are not necessarily identical under hypoxic conditions. It has been suggested that *Hox* genes may modulate the adaptation potential of stem cells to environmental signals from the microenvironment and surrounding tissues. The absence of *Hox* gene expression in embryonic stem cells (ESCs) implies that a less stringent *Hox* expression status in stem cells may afford them additional plasticity, particularly in response to environmental conditions [2,20,21,22]. The tissue-specific pattern of *Hox* gene expression, known as the *Hox* code, is maintained in adult cells, such as fibroblasts, MSCs, and osteoprogenitor cells. Several in vivo and in vitro studies have reported on the direct contribution of *Hox* genes in differentiation [23,24,25,26].

Furthermore, the degree to which *Hox* expression profiles differ in MSCs depends on their source of isolation, and this pattern determines their differentiation potential. MSCs originating from the neuroectoderm can be distinguished from their mesodermal counterparts by their *Hox*-negative profile. MSCs with negative *Hox* gene expression have been shown to possess higher proliferation and differentiation capacities, including performing chondrogenesis, which are favorable characteristics in cell therapy. An intriguing hypothesis is the dependency of MSCs’ plasticity on the *Hox* expression pattern, where *Hox*-negative MSCs respond more robustly to environmental factors compared to *Hox*-positive cells [27,28,29].

Neural crest multipotent cells are exclusive to vertebrates and typically do not express *Hox* genes. In addition to their role in early embryonic development, these cells are present in various neural-crest-derived tissues in fetal and adult organisms, and they include DPSCs [27,30,31].

In this study, the chondrogenic differentiation of human BMSCs with a mesodermal origin and a *Hox*-positive expression pattern was compared to human DPSCs, with the reported negative expression of some *Hox* genes. The plasticity of each cell source was evaluated based on the level of altered chondrogenic differentiation efficiency in response to hypoxia-mimicking conditions.

## 2. Materials and Methods

### 2.1. Stem Cell Cultures

The human DPSCs and BMSCs were provided by the Stem Cells Technology Research Center (Shiraz, Iran) and Bonyakhteh Research Center (Tehran, Iran), respectively. Both cell types had been obtained from healthy donors (male and female) aged 19 to 32 years. The cells were cultured in Dulbecco’s Modified Eagle Medium: Nutrient Mixture F-12 (DMEM/F-12) (BioIdea Co., Tehran, Iran) and supplemented with 10% Fetal Bovine Serum (FBS) (Gibco Co., Waltham, MA, USA) and 1% penicillin/streptomycin (Gibco Co.). The culture medium was refreshed every 3 days. The cultures were maintained in a humidified incubator at 37 °C with 5% CO_2_, and the MSCs in passages 4–6 were harvested for the experiments at 70–80% confluency.

### 2.2. Cell Viability Assays

The hypoxia preconditioning was conducted using CoCl_2_ (Merck Co., Munich, Germany). The following four experimental groups were included: untreated and CoCl_2_-pretreated DPSCs and untreated and CoCl_2_-pretreated BMSCs. The CoCl_2_ was dissolved in distilled water, and final concentrations of 100, 250, 500, and 1000 µM were applied to treat the MSCs for 24 h. Concentrations with no significant harmful effects were chosen to treat the MSCs at different exposure times (3, 7, and 10 days) to determine the optimum treatment with minimal adverse effects. A colorimetric MTT assay was utilized to evaluate the effect of the CoCl_2_ on the stem cells. In this test, water-soluble 3-(4,5-dimethylthiazol-2-yl)-2,5-diphenyltetrazolium bromide salt (Carl Roth Co., Karlsruhe, Germany) was metabolized by cultured viable cells to a purple, water-insoluble formazan. After dissolving the formazan in dimethyl sulfoxide (DMSO) (Merck Co.), the optical density (OD) at 570 nm was measured with a multi-well plate reader (Mikura Co., York, UK).

### 2.3. Three-Dimensional Pellet Cultures and Chondrogenic Differentiation

Following harvesting the cells, 2.5 × 10^5^ MSCs in 250 µL of chondrogenic media (StemPro^®^ Chondrogenesis Differentiation Kit, Gibco Co.) was transferred to screw-cap polypropylene conical tubes (Extragene Co., Taichung, Taiwan). The MSCs were then centrifuged (200× *g*, 10 min, 25 °C) to form cell pellets at the bottoms of the conical tubes. The chondrogenic media were gently refreshed every 3 days over a period of 3 weeks.

### 2.4. Morphological and Histological Experiments

#### 2.4.1. Pellet Morphology Assessment

At the end of the differentiation period, the pellet condensation and integrity were surveyed qualitatively by compressing the pellets with a pipettor’s tip. The gross pellet morphologies, including their sizes, colors, and elastic states, were compared to each other.

#### 2.4.2. Histochemical Assessments

The formalin-fixed chondrogenic pellets were embedded in paraffin (Sigma Co., Munich, Germany). Briefly, the preparation steps for the pellet samples were conducted using an automatic tissue processor in the following order: dehydration, clearing, embedding, and blocking. The processed paraffin-embedded pellets were cut by a microtome into 3 μm sections. The ECM structures were qualitatively validated with hematoxylin (Sigma Co.) and eosin (Sigma Co.) (H&E) staining, and alcian blue (Sigma Co.) staining for the evaluation of GAGs deposition was performed. Briefly, the hematoxylin and eosin staining protocol involved deparaffinizing and rehydrating the slides to water, followed by staining them with hematoxylin and washing them in tap water. Subsequently, the slides were passed through 0.3% acid alcohol, rinsed in tap water, and stained with 1% eosin. Dehydration was then carried out using 70%, 90%, and 100% ethanol, followed by clearing the slides in xylene. Finally, the slides were mounted.

For the alcian blue staining, slides were deparaffinized and hydrated to distilled water. They were then stained in alcian blue solution for 30 min, followed by a wash in running tap water and a rinse in distilled water. The nuclei were counter-stained with nuclear fast red solution, washed in running tap water, and rinsed in distilled water. Sections were dehydrated using 95% ethanol and absolute ethanol. After clearing in xylene, the slides were mounted with a mounting medium.

#### 2.4.3. Immunohistochemical Experiments

The immunohistochemical assessment was conducted to detect collagen II and collagen X levels to evaluate the production of chondrogenic and hypertrophic markers, respectively. Primary antibodies including rabbit anti-collagen II polyclonal (Abcam Co., Cambridge, UK) and rabbit anti-collagen X polyclonal (Abcam Co.) were utilized, and their detection was achieved through using horseradish peroxidase (HRP)-conjugated goat anti-rabbit IgG (Abcam Co.) as the secondary antibody.

The procedure involved deparaffinizing the sections followed by rehydration using a descending ethanol dilution series. Antigen retrieval was carried out through heat-induced epitope retrieval. Endogenous peroxidase activity was quenched using a 3% H_2_O_2_ diluted in methanol. To block non-specific binding sites, a blocking solution (300 µL phosphate buffered saline (PBS) containing 10% goat serum (Abcam Co.) and 5% bovine serum albumin (BSA) (Sigma Co.)) was applied, after which the sections were incubated with the primary antibodies. Subsequently, HRP-conjugated secondary antibody was used. Color development was achieved using diaminobenzidine tetrahydrochloride (DAB) (Sigma Co.). The sections were stained with hematoxylin, dehydrated using an ethanol series, and mounted with Entellan mounting medium (Sigma Co.). Images were captured with a Nikon eclipse E200 microscope.

#### 2.4.4. Estimating the Volume Densities of the Chondrogenic Pellets Components

To determine the volume densities of the components within the chondrogenic pellets, the stained sections were analyzed using a video-microscopy system (Nikon Co., Waltham, MA, USA). Approximately 5 microscopic fields in each section were systematically examined and scanned along the X and Y axes using a stage micrometer. A point probe with 96 points was then positioned over each sample section’s image as displayed on the monitor. Subsequently, the volume density (Vv) of each chondrogenic component was evaluated using the point counting method and the following equation:V_V(*Collagen* II/X)_ = ΣP_(*Collagen* II/X)_/ΣP_(*Nuclei*),_
where ΣP_(*Collagen* II/X)_ and ΣP_(*Nuclei*)_ represent the total points in each component’s profile of collagens and the nuclei of the cells, respectively.

### 2.5. Biochemical Analysis

#### 2.5.1. ICP Mass Spectrophotometry

Cell culture media were sampled before and after 24 h treatment of the MSCs with 100 μM of CoCl_2_. Quantitative measurements of the cobalt in the culture media were conducted using inductively coupled plasma mass spectrophotometry (ICP-MS) (Varian Vista-Pro, Santa Clara, CA, USA). The amount of cobalt uptake was then normalized to the cell number.

#### 2.5.2. Cell Lysate Preparation

To prepare the cell lysate, 3D pellets were harvested, washed with PBS, and mechanically homogenized in 0.3 mL of alkaline lysis buffer. Subsequently, the homogenate was incubated for one hour at 25 °C with intermittent vortexing. After centrifugation at 10,000× *g* at 4 °C for 20 min, the supernatants were aspirated and aliquoted to new tubes on ice. The microtubes containing the lysates were then stored at −80 °C until further use for the ALP specific activity assay and protein quantification.

#### 2.5.3. Protein Quantification

The protein concentrations in the pellet lysates were measured using a Pierce bicinchoninic acid (BCA) assay kit (Thermo Fisher Scientific Co., Waltham, MA, USA) with BSA as the standard, according to the kit instructions. The absorbance of the purple-colored product was measured at 562 nm, and the protein concentration for each sample was determined using the calibration curve of the BSA standards.

#### 2.5.4. ALP Specific Activity Assay

In the ALP specific activity assay, the colorless phosphatase substrate, p-nitrophenyl phosphate (pNPP) (Sigma Co.), was hydrolyzed to a yellow-colored end product, pNitrophenol, after administration on the cell lysate. The absorbance of the yellow product at 405 nm was proportional to the ALP activity, which was measured using the following formula (in which 18.45 is the extinction coefficient (ε), R is the dilution factor divided by the path length, and t_x_ is 30 min):ALP activity (Units/mL) = ((A_final_ − A_initial_)/t_x_) × R/18.45.

The ALP activity was then normalized to the protein content of each sample and reported as the specific ALP activity (mU/mg).

#### 2.5.5. DNA Quantification

The DNA in each pellet was quantified using an AccuPrep Genomic DNA extraction kit (BioNeer Co., Oakland, CA, USA), following the manufacturer’s instructions. The DNA yield (ng/μL) was determined by measuring the OD at 260 nm using a NanoDrop spectrophotometer, and the concentration of each sample was then normalized to its volume.

#### 2.5.6. GAGs Quantification

To fix the pellets, −20 °C methanol (Sigma Co.) was used for 10 min. After aspirating the methanol and washing it, the GAGs staining was performed using 1% alcian blue solution overnight. The next day, the alcian blue was washed by distilled water, and then afterward, 150 µL guanidine hydrochloride (GuHCl) (Sigma Co.) 6 M in distilled water was added to the tubes and incubated at room temperature for 5 h. The OD of the extracted alcian blue stain was measured at 630 nm, which was proportional to the GAGs content of the 3D pellets. The DNA content of the corresponding group was used to normalize the results.

### 2.6. Molecular Experiments

#### 2.6.1. RNA Extraction and Reverse Transcription–Polymerase Chain Reaction (RT-PCR)

The total RNA was extracted from the stem cells using Biozol reagent (Bioflux Co., Tokyo, Japan), and RNA quantification was evaluated using a NanoDrop 1000 spectrophotometer (Thermo Fisher Scientific Co., Waltham, MA, USA). The genomic DNA was removed through DNase (Thermo Fisher Scientific) treatment, and cDNA synthesis was performed according to the guidelines provided by the Revert Aid First Strand cDNA Synthesis Kit (Thermo Fisher Scientific). Additionally, the primers were designed using AlleleID 7 software, and their specificities were verified using Primer-BLAST. All primers were procured from Takapouzist, Tehran, Iran, and their sequences are listed in Appendix A.

#### 2.6.2. Quantitative Real-Time PCR

Quantitative real-time PCR (qPCR) was used to assess the gene expression for *Hox A5*, *Hox A7*, and *Hox C10* with SYBR green master mix (SYBR^®^ Premix Ex Taq ™ II, Takara Co., Kusatsu, Japan). The PCR amplifications were carried out using an ABI real-time PCR 7500 system. The qPCR program is detailed in Appendix A. The efficiency of the amplification was evaluated using the slope of a standard curve derived from the tenfold serial dilutions of the pooled cDNA. Data analysis was performed using 7500 software v2.0.1. The Ct values obtained for the *Hox* genes were normalized against that of β-actin, and the relative gene expression was represented as the fold change (2^−∆Ct^). The differences in the Ct values (delta Ct) of the target gene was calculated, and the data were presented as percentages or fold differences in comparison to the control.

### 2.7. Statistical Analysis

All data resulted from at least three independent experiments. The Kolmogorov–Smirnov test was applied for the normality assessment of the quantitative data. SPSS 16 software was employed for the statistical analysis using an independent samples *t*-test and one-way and two-way ANOVA with a Tukey post hoc test. GraphPad Prism 8.0 was utilized to generate the graphs. The data are presented as means ± SEMs, and differences with *p*-values of <0.05 were statistically considered significant.

## 3. Results

### 3.1. MSCs Culture and Morphology

The spindle-shaped passages 4–6 of the DPSCs and BMSCs at 80% confluency were observed in a monolayer culture (Figure 1).

#### 3.1.1. Cell Viability

The treatment with CoCl_2_ at a concentration of 100 μM for 24 h did not adversely affect the viability of the stem cells. However, the extended exposure of the cells to 100 μM of CoCl_2_ for 3 days or more had a significant adverse effect on the DPSCs (*p* < 0.001). Moreover, CoCl_2_ at a concentration of 250 μM resulted in a significant decrease in the DPSCs’ viability (*p* < 0.001) (Figure 2).

Notably, variations were observed between the DPSCs and BMSCs in terms of viability following the CoCl_2_ treatments. CoCl_2_ at both the 100 and 250 μM concentrations did not exert significant effects on the BMSCs, but concentrations higher than 250 μM adversely affected their viability after 24 h (*p* < 0.001) (Figure 2). Additionally, prolonged exposure of the BMSCs to 100 μM CoCl_2_ did not significantly affect cell viability for 7 days, while 250 μM of CoCl_2_ for 7 days or more resulted in decreased cell viability (*p* < 0.001). Based on the MTT results for both stem cell sources, treatment with 100 μM of CoCl_2_ for 24 h was selected to mimic hypoxia before inducing the cells toward chondrocyte differentiation.

#### 3.1.2. Chondrogenic Differentiation

Pellets were successfully formed at the bottoms of the conical tubes, and on day 21 of the chondrogenic induction, they were harvested to compare the chondrogenic differentiation efficacy of the pretreated and untreated BMSCs and DPSCs.

### 3.2. Morphological and Histological Experiments

#### 3.2.1. Pellet Morphology

The macroscopic assessment revealed that the pellets derived from DPSCs developed firmer structures earlier than their BMSCs counterparts. However, by day 21 of differentiation, both groups exhibited white, round shapes with elastic textures (Figure 3). Notably, integrated, round-shaped structures began forming by day 5 in the CoCl_2_-pretreated DPSCs, whereas it took approximately 10 days for these to develop in the untreated cells cultured in the conical tubes. A similar difference in the formation of integrated structures was observed in the BMSC-derived pellets. The pellet sizes were remarkably larger in all CoCl_2_-preconditioned DPSCs than in the untreated pellets, though this difference was less noticeable in the case of the BMSCs chondrogenic pellets.

#### 3.2.2. Hematoxylin and Eosin Staining

The hematoxylin and eosin staining of the MSC chondrogenic pellet sections was applied to evaluate the tissue-like structures. The pellet sizes were notably greater in the preconditioned DPSCs, displaying more rounded and regular shapes. This difference was less considerable in the BMSC samples. The histochemical assessment also unveiled that lacuna-like structures were well organized in the CoCl_2_-pretreated pellets compared to the untreated samples. These structures were superiorly created in the BMSCs compared to the DPSCs chondrogenic pellets (Figure 4).

#### 3.2.3. Alcian Blue Staining of the MSC Chondrogenic Pellet Sections

Alcian blue staining was employed to confirm the chondrogenic differentiation. This staining in the pellets differentiated from the DPSCs unveiled a significantly higher deposition of GAGs in the extracellular matrix of the CoCl_2_-preconditioned samples compared to the untreated pellets. Overall, in the pellets differentiated from the BMSCs, the GAGs deposition was comparable between the treated and untreated samples (Figure 5).

#### 3.2.4. Immunohistochemical Assessments

The expression of collagen II was higher in the CoCl_2_-pretreated pellets derived from the DPSCs compared to the untreated samples, as demonstrated by the immunohistochemistry assay. Additionally, in the pellets derived from the BMSCs, the expression of collagen II also increased due to the preconditioning. The expression of collagen II in the treated and untreated samples differentiated from the BMSCs was higher compared to the DPSCs samples (Figure 6).

The CoCl_2_ pretreatment did not alter the collagen X production in the DPSCs samples significantly, while in the pellets differentiated from the BMSCs, the CoCl_2_ pretreatment unfavorably increased the expression of collagen X compared to the untreated samples. Overall, the production of collagen X was higher in both the untreated and treated BMSCs samples compared to their DPSCs counterparts (Figure 7).

#### 3.2.5. Quantitative Analysis of Collagen II and X Volume Densities in the Chondrogenic Pellets

The results demonstrated a significant increase of 2.3-fold and 84% in the volume density of collagen II in the pretreated DPSCs and BMSCs samples compared to their untreated counterparts, respectively (*p* < 0.0001). The untreated BMSCs pellets showed significantly higher volume densities of collagen II compared to the untreated DPSCs pellets (*p* < 0.01), while the differences were not significant for the treated DPSCs and BMSCs samples (Figure 8).

The analyzed data obtained from the volume densities of collagen X in the treated DPSC samples showed decreases compared to the untreated samples, but the decreases were not significant. In contrast, the volume densities in the treated BMSCs pellets showed a significant increase of 28% compared to the untreated pellets (*p* < 0.05). The untreated BMSCs samples exhibited significantly higher volume densities of collagen X compared to the untreated DPSCs samples (*p* < 0.01), and the differences were significant in the treated BMSCs and DPSCs samples in the same manner (*p* < 0.0001) (Figure 8).

### 3.3. Biochemical Analysis

#### 3.3.1. CoCl_2_ Uptake

The analysis using ICP-MS revealed that the DPSCs exhibited a significant lower uptake of CoCl_2_ from the culture medium compared to the BMSCs (*p* < 0.01) (Figure 9).

#### 3.3.2. GAGs Deposition

The further analyses of the alcian blue eluted from the stained pellets showed significantly elevated GAGs contents in the CoCl_2_-pretreated DPSCs pellets compared to the untreated DPSCs pellets (*p* < 0.001), while the preconditioning of the BMSCs samples did not significantly affect the GAGs contents (Figure 10). The GAGs deposition was significantly higher in the untreated samples derived from the BMSCs compared to the untreated pellets derived from the DPSCs (*p* < 0.05). However, the GAGs deposition in the treated samples derived from the BMSCs was not significantly different compared to the CoCl_2_-treated pellets derived from the DPSCs.

#### 3.3.3. ALP Specific Activity

To evaluate the tendency of the differentiated pellets to mineralization, the ALP specific activity was measured. The results showed that the CoCl_2_ pretreatment had no significant effect on the ALP specific activity in the differentiated BMSCs and DPSCs. However, the ALP specific activity was significantly higher in the untreated and treated pellets differentiated from the BMSCs compared to their DPSCs counterparts, with negligible activity (Figure 11) (*p* < 0.001).

### 3.4. Molecular Experiments

#### Real-Time PCR

The results showed that the mRNA for Hox A5, Hox A7, and Hox C10 were expressed in the BMSCs, while no expression of these genes was detected in the DPSCs (Figure 12).

## 4. Discussion

The application of MSCs, particularly autologous MSCs, may potentially make cartilage repair more widely available [2,32]. However, there are still some uncertainties regarding the most appropriate cell source for cartilage regeneration. BMSCs are the most studied and clinically used type of MSC. However, they have failed to be adequately expanded ex vivo, and their harvesting procedure is invasive [33,34]. Meanwhile, DPSCs have been introduced as easily accessible multipotent stem cells with higher expansion potential compared to BMSCs for meeting the demand for a high number of cells in cell transplantation [35,36].

There is a paucity of research on the chondrogenic differentiation of DPSCs, and their exhibited limited expression of collagen II and production of GAGs was consistent with our results, and this impedes their applications for cartilage cell therapy [37]. On the other hand, the favorable characteristics of DPSCs encourage further endeavors to use these cells as a reasonable source in cell therapy.

During our experiments, the DPSCs were successfully sub-cultured at least ten times without the loss of their normal morphology or reduced proliferation compared to the BMSCs, which showed deformed morphology and failed to proliferate efficiently after seven passages. The greater expansion capacity of DPSCs has been mentioned in previous studies, in agreement with our results [36,38].

Micromass or pellet culture, which was applied in the current study, offers 3D, closely packed cellular aggregates, allowing cell–cell interactions like the pre-cartilage condensation stage. Cartilage development initiates with the condensation of mesenchymal progenitor cells, which is an essential step in the chondrogenesis pathway [39,40]. Notably, the 3D culture of chondrocytes has been shown to maintain their articular character and prevent the de-differentiation typically observed in monolayer cultures [41]. Drawing on these concepts, a two-step, directed chondrogenic induction protocol was optimized, and it included a preconditioning step followed by a pellet culture step. Pretreatment of the cells with CoCl_2_ necessitated a cell viability assay. Inconsistent with some reports, our results showed that the effect of CoCl_2_ on cell viability is dose- and time-dependent [33,42].

Chondrogenic differentiation of MSCs is usually confirmed by morphological assessments such as pellet size, hyaline cartilage ECM deposition, and hypertrophic markers via histochemical and immunohistochemical methods, the expression of transcription factors, DNA content, etc. [43]. The pretreated DPSC pellets showed the largest sizes after 3 weeks and were bulkier compared to the untreated samples. This observation may be attributed to the increased deposition of ECM molecules under the hypoxia-mimicking conditions. It was reflected in greater spacing between the cell nuclei compared to the condensed and cell-populated routine cultured pellets with a low matrix production, as demonstrated in the hematoxylin-and-eosin-stained sections. This finding was in accordance with previous studies that reported enhanced ECM deposition and reduced cell density by hypoxia and hypoxia-mimicking agents [42,43,44]. Cells enclosed in cartilage ECM acquire a round shape, which is a specific morphological characteristic of a chondrocyte, and this was observed in our differentiated pellets. Moreover, the specific cartilage lacuna-like structures were more prominent in the CoCl_2_-preconditioned pellets, indicating the positive effect of CoCl_2_ pretreatment. This observation aligned with previous studies which demonstrated that transient hypoxia could favorably lead to the formation of lacuna spaces in a cartilaginous matrix [44,45].

The earlier observed firmness and larger sizes of the pellets derived from the DPSCs compared to the BMSCs pellets could be interpreted as being due to the higher responsiveness of the DPSCs to the environmental factor, i.e., the CoCl_2_.

The enhanced expression of collagen II and the GAGs contents in response to the CoCl_2_ preconditioning indicated improved chondrogenic differentiation in the DPSCs. Similarly, collagen II levels increased in the BMSCs after the preconditioning; however, the difference in GAGs deposition between the pretreated and untreated BMSCs samples was not significant. This could be interpreted as the higher plasticity of the DPSCs in response to the CoCl_2_ compared to the BMSCs. The non-significant differences between the collagen II levels and GAGs depositions of the preconditioned DPSCs and BMSCs indicated the hypoxia-mimicking potential to compensate for the lower ability of the DPSCs in producing chondrogenic markers.

Early core cartilage enlarges by the proliferation of chondrocytes and the production of a matrix. Subsequently, cartilage formation progresses through the process of maturation and hypertrophy, along with the production of collagen X [46]. Hypertrophic chondrocytes facilitate matrix mineralization, recruit vascular and bone cells, and, eventually, undergo cell loss via apoptotic cell death. In fact, collagen X is the marker for cartilage hypertrophy and endochondral ossification, which serves as a calcification framework [47,48].

Unlike the DPSCs, the pellets derived from BMSCs unfavorably produced greater amounts of collagen X in response to the CoCl_2_. Some controversial results have been reported regarding the effect of hypoxia induction on collagen X expression [49,50,51].

Another concern in in vitro MSC chondrogenesis is the induction of ALP activity as an early osteogenic marker [52,53,54]. In our study, no significant change in ALP activity was detected in response to the hypoxia-mimicking agent. What was noticeable in our results was that the ALP specific activity was more than six-fold higher in the BMSCs-derived samples compared to the DPSCs-derived samples. This could imply a desirable parameter for the application of DPSCs as an alternative MSC cell source in cartilage tissue engineering due to their lower propensity toward hypertrophy, which would otherwise result in apoptosis and ossification [55].

Some studies have highlighted the intriguing effects of hypoxia on MSC function, including their secretome composition, which might also be involved in the chondrogenic differentiation effects of hypoxia-mimicking agents [2,18,56].

An explanation for the current results could be the referred of HIF-1α, an oxygen-sensitive transcription factor, as its activity is induced in response to low oxygen tension and hypoxia-mimicking agents. Numerous genes are regulated by this transcription factor [18]. HIF-1α exerts its critical role in cartilage development in part through regulating Sox9, a chondrogenic marker gene which is required as a transcription factor for the initiation of chondrogenesis. The synthesis of cartilage-specific ECM rich in GAGs and collagen II is preceded by the expression of sox9. It may also ameliorate the hyperthrophic propensity of MSCs during chondrogenesis, followed by a loss of cells through apoptosis [2].

To explore whether the different responses to the hypoxia-mimicking agent in cell viability and chondrogenic differentiation efficiency resulted from the unequal uptake of CoCl_2_ by two stem cell sources, the amount of the reduction in cobalt ions from the media after 24 h was measured by ICP-MS. It was found that, despite their superior response to the CoCl_2_, the DPSCs showed a lower uptake of cobalt compared to the BMSCs, which could be attributed to the intrinsic properties of these cells, including their plasticity. Our results highlight that despite the identical concentrations of the CoCl_2_ in the culture media, the amounts of the CoCl_2_ in each cell type did not necessarily correspond to the administered concentrations. Given the ongoing research on hypoxia mimicking in various fields of study, from regenerative medicine to cancer, it might be advisable for future studies to consider the available concentrations of CoCl_2_ inside the cells rather than just the administered concentrations in the cell culture medium.

The expression profiles of the *Hox* genes could be considered as parameters for the selection of an MSC source for cell therapy. More efficient chondrogenic differentiation with lower hypertrophy in DPSCs pellets supports the hypothesis that *Hox*-negative MSCs possess higher plasticity than BMSCs, which express *Hox* genes. However, the *Hox* expression pattern of DPSCs has not been completely studied, and the only report refers to Pelttari et al. [27], who demonstrated negative expression for *HoxC4*, *Hox C5*, *Hox C8*, *Hox D3*, and *Hox D8*. In agreement with the reported *Hox*-negative expression patterns of neural-crest-derived cells, the qPCR results showed that the DPSCs did not express *Hox A5*, *Hox A7*, and *HoxC10,* which could be added to the previously studied *Hox* gene expression profile.

For BMSCs, consistent with our results, Liedtke et al. described the positive expression of these *Hox* genes by RT-PCR [29]. In future regenerative medicine, the application of *Hox* gene expression profiles for the precise selection of stem cell types in different cell therapies and the manipulation of *Hox* gene plasticity will likely become an active area of research.

The two-step directed chondrogenic induction protocol for DPSCs, as an almost unlimited cell source, represents a simple and practical approach for chondrogenic differentiation, leveraging the potential of *Hox*-negative cells and offering a cost-effective alternative for cartilage regenerative medicine.

## 5. Conclusions

This study attempted to advise of a molecular criterion for selecting a suitable cell source for chondrogenic differentiation and cartilage cell therapy. While further research is necessary, our initial findings unveiled that *Hox* genes have the potential to serve as valuable molecular markers for discriminating between different cell sources to coordinate with cell therapies.

DPSCs exhibit favorable properties, including a minimally invasive isolation procedure and high proliferation capacity. These cells have demonstrated the potential for advantageous modification of their less desirable traits, such as limited chondrogenic differentiation, in response to a hypoxia-mimicking agent, partially attributed to their negative *Hox* code.

Moreover, it is worth noting that, for the first time, a diminished tendency of DPSCs toward hypertrophy was observed, thereby enhancing their proportion for application in cartilage cell therapy.

Given the known deficiencies of MSC-based cartilage cell therapies, such as cell loss during differentiation, limited size of cartilage-like tissues, and undesired overexpression of hypertrophic factors, our data emphasize the potential impact of using a hypoxia-mimicking strategy for DPSCs chondrogenic differentiation.

## Figures and Tables

**Figure 1 biology-13-00560-f001:**
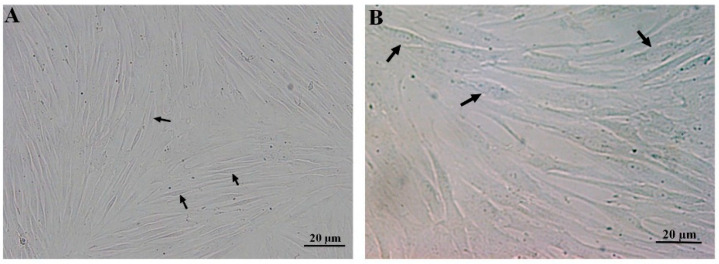
Fibroblast-like morphology of passages 4–6 of the DPSCs (**A**) and BMSCs (**B**) in a monolayer culture, forming swirls at high confluency. Each arrow shows a representative stem cell.

**Figure 2 biology-13-00560-f002:**
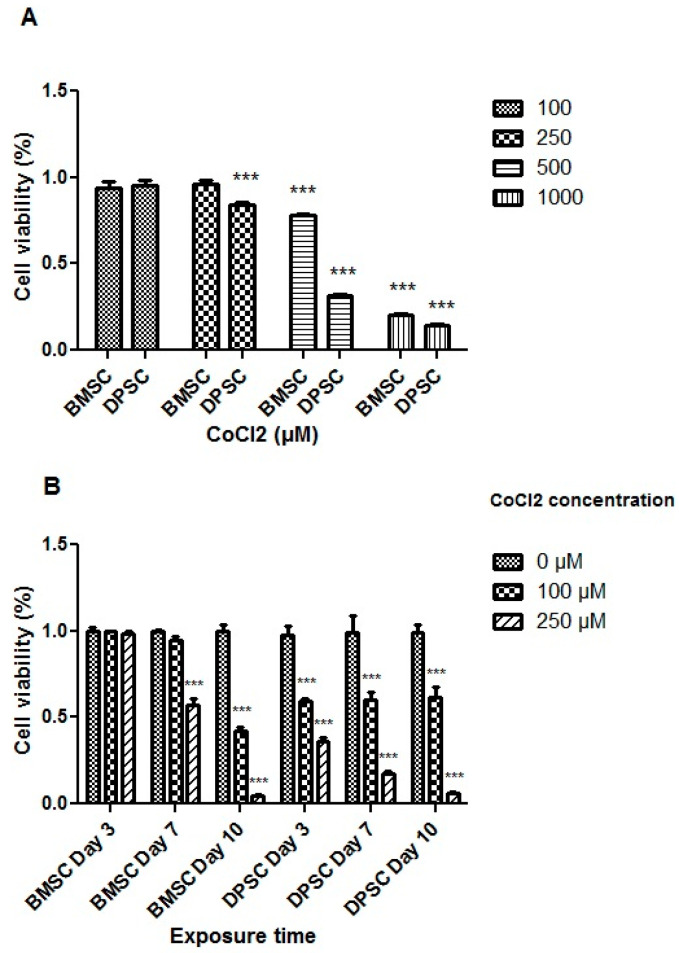
Cell viability assay for the different CoCl_2_ concentrations (**A**) and exposure times (**B**). All treatments were compared to the control groups and represented as percentages of viability. ***, *p* < 0.001.

**Figure 3 biology-13-00560-f003:**
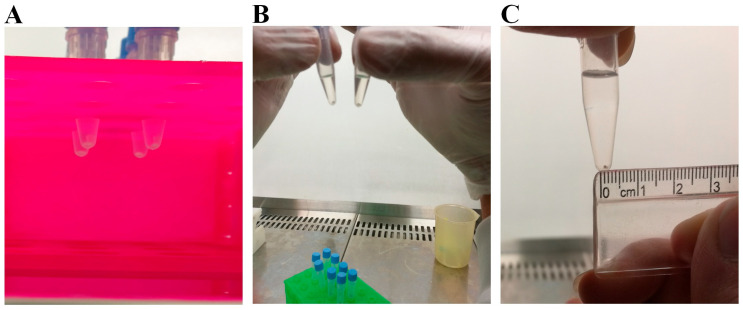
Gross morphology of the chondrogenic pellets at the initiation of differentiation (**A**), the round morphology at the end of differentiation (**B**), and the approximate size (**C**).

**Figure 4 biology-13-00560-f004:**
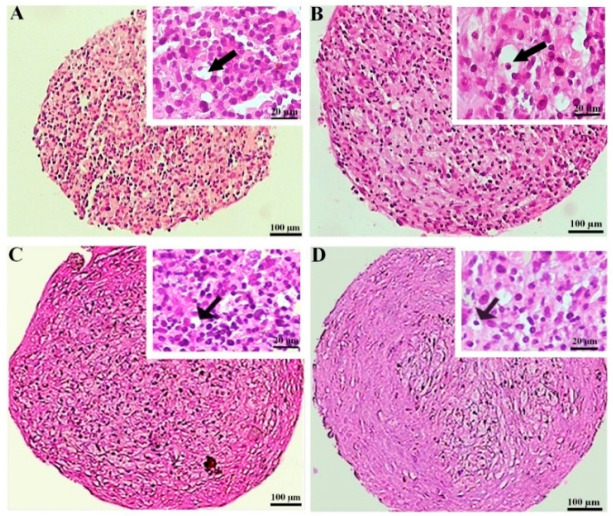
Tissue structures of the pellets stained by hematoxylin and eosin at day 21 of differentiation in the untreated (**A**,**C**) and CoCl_2_-pretreated (**B**,**D**) pellets derived from the DPSCs (**A**,**B**) and BMSCs (**C**,**D**). The lacuna-like structures are shown with black arrows.

**Figure 5 biology-13-00560-f005:**
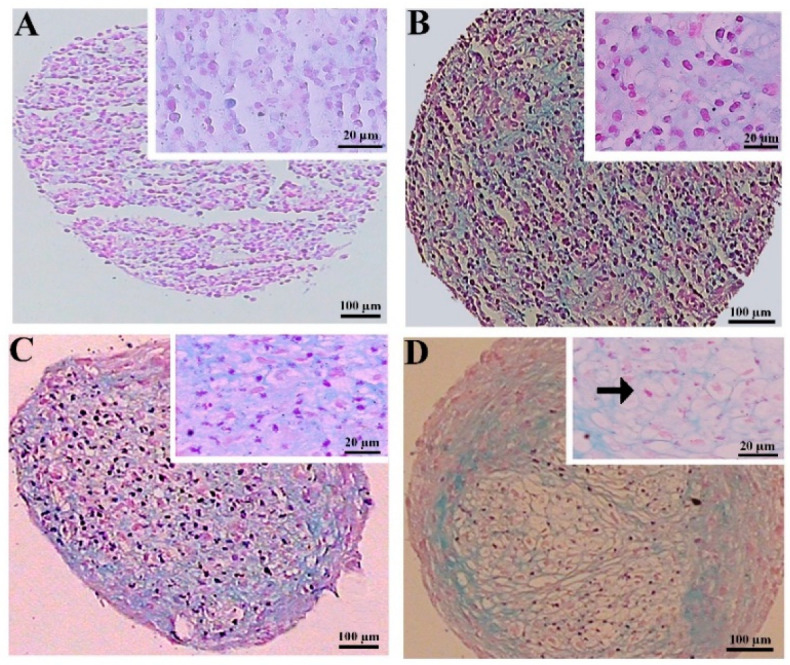
GAGs staining by alcian blue at day 21 of differentiation in the untreated (**A**,**C**) and CoCl_2_-pretreated (**B**,**D**) samples for the DPSCs (**A**,**B**) and BMSCs chondrogenic pellets (**C**,**D**). A lacuna-like structure is shown with a black arrow.

**Figure 6 biology-13-00560-f006:**
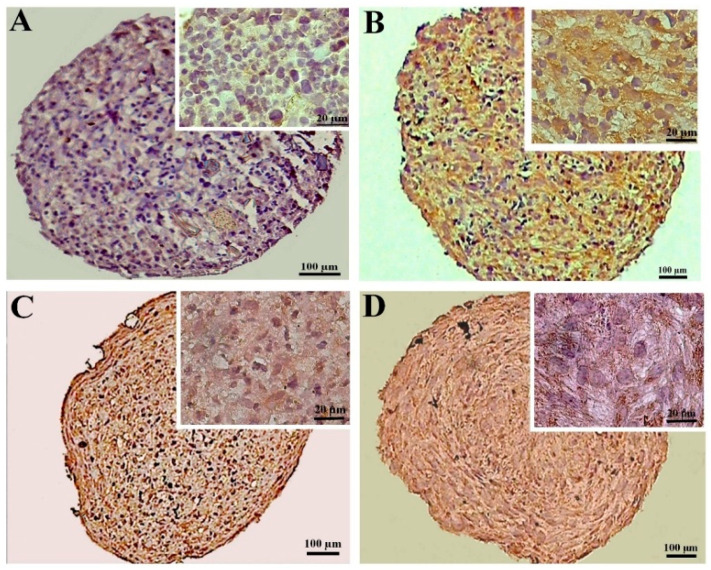
Immunostaining for collagen II at day 21 of chondrogenic induction in the untreated (**A**,**C**) and CoCl_2_-pretreated (**B**,**D**) pellets derived from the DPSCs (**A**,**B**) and BMSCs (**C**,**D**).

**Figure 7 biology-13-00560-f007:**
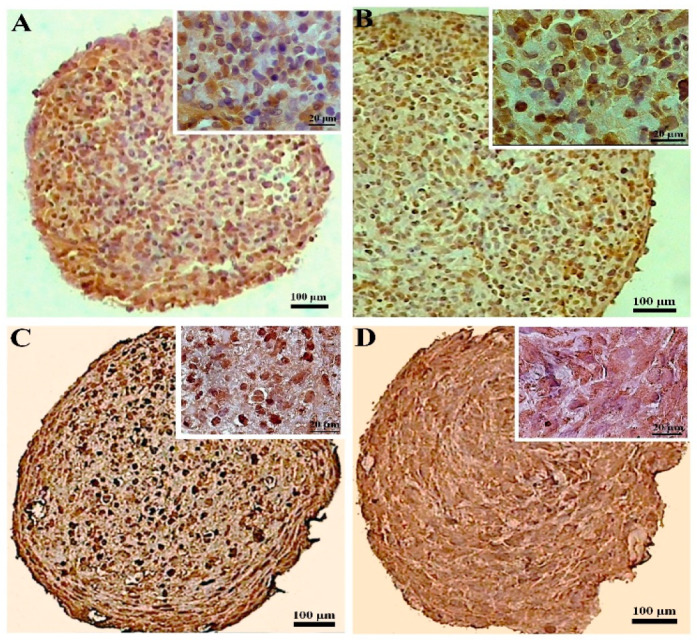
Immunostaining for collagen X at day 21 of chondrogenic induction in the untreated (**A**,**C**) and CoCl_2_-pretreated (**B**,**D**) pellets derived from the DPSCs (**A**,**B**) and BMSCs (**C**,**D**).

**Figure 8 biology-13-00560-f008:**
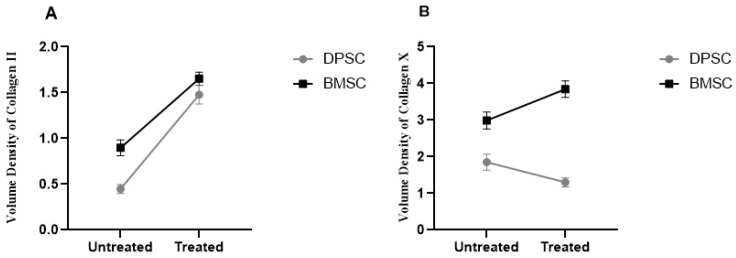
Volume densities of collagen II (**A**) and collagen X (**B**) in the untreated and CoCl_2_-pretreated pellets.

**Figure 9 biology-13-00560-f009:**
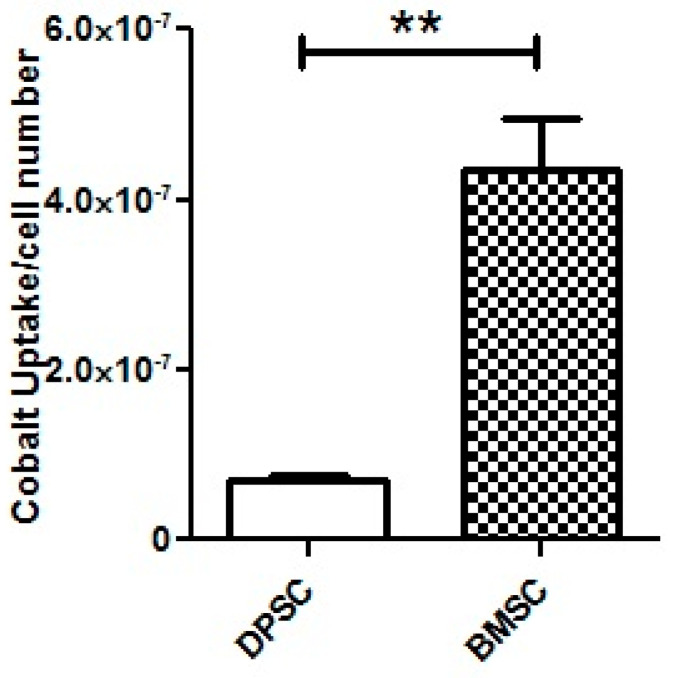
ICP-MS for the assessment of CoCl_2_ uptake by the DPSCs and BMSCs after 100 μM of CoCl_2_ treatment for 24 h. **, *p* < 0.01.

**Figure 10 biology-13-00560-f010:**
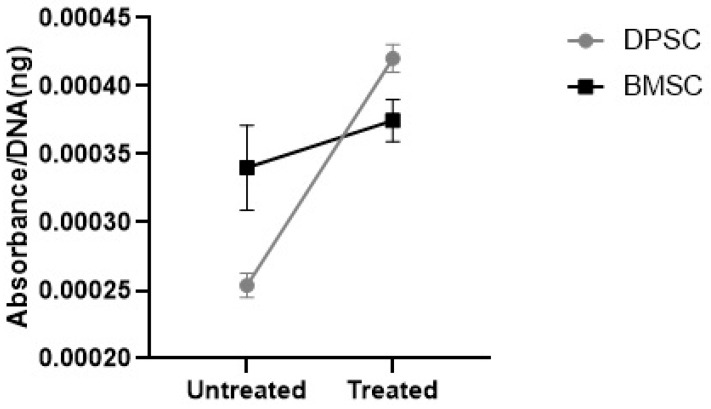
The GAGs contents were shown as the absorbance of the eluted alcian blue compared to the DNA content (ng) ratio.

**Figure 11 biology-13-00560-f011:**
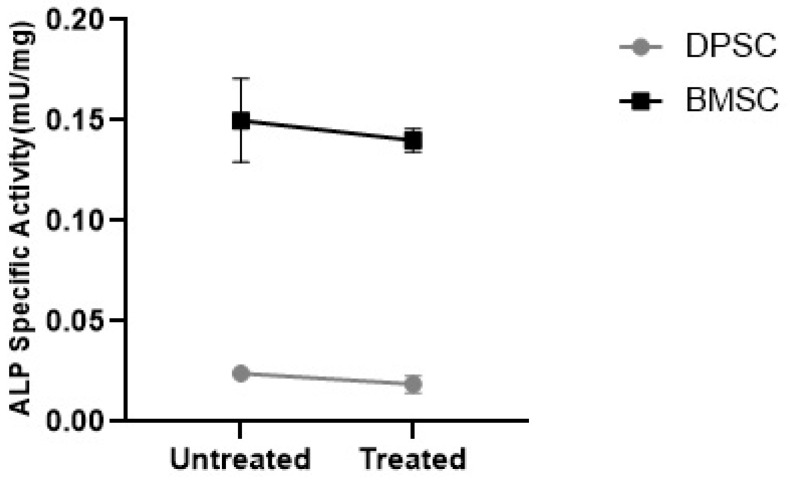
ALP specific activity in the MSC chondrogenic pellets represented as mU/mg.

**Figure 12 biology-13-00560-f012:**
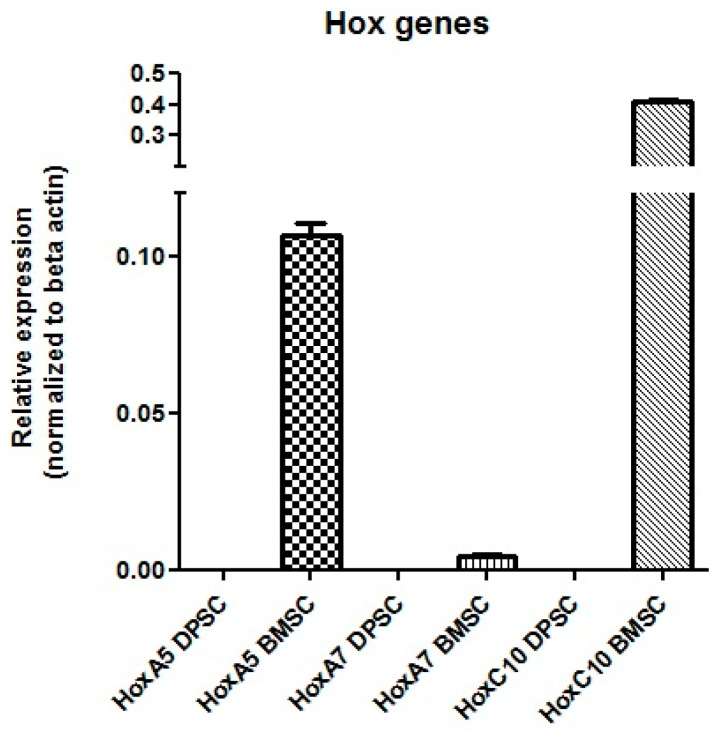
mRNA expression of the Hox genes in the DPSCs and BMSCs.

## Data Availability

The original contributions presented in the study are included in the article/Appendix A, further inquiries can be directed to the corresponding authors.

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
