# Peer review of "Plasticity Comparison of Two Stem Cell Sources with Different Hox Gene Expression Profiles in Response to Cobalt Chloride Treatment during Chondrogenic Differentiation"

_biology, 2024, doi:10.3390/biology13080560_

Round 1
Reviewer 1 Report
Comments and Suggestions for Authors
Review comments
“Plasticity Comparison of Two Stem Cell Sources with Different 2 Hox Gene
Expression Profiles in Response to Cobalt Chloride 3 Treatment During
Chondrogenic Differentiation.”
1. The data presented in Fig. 5 and 6 regarding collagen Ⅱ and Ⅹ expression is
important for understanding the differential response of DPSCs and BMSCs to
hypoxic conditions. Including graphs that quantitatively depict these results can
provide a visual representation of the differences in collagen expression level,
making it easier to interpret the data.
2. The introduction provides a comprehensive overview of the challenges in cartilage
regeneration and the potential of mesenchymal stem cells. However, it would be
beneficial to provide more detailed information on the role of hypoxia in stem cell
differentiation.
3. Your study highlights the differences in chondrogenic differentiation between
DPSCs and BMSCs with different HOX gene expression profiles. In the
introduction, it would be useful to include more detailed information on the role of
HOX gene in stem cell, for example, in stem cell behavior, differentiation potential,
plasticity, and responsiveness to environmental factors.
4. The explanation of Hox gene within the Introduction part seems insufficient.
5. In Fig. 1, what does the large size of BMSCs signify?
6. Does CoCl2 play a role during pellet generation after pretreatment, and was there
no comparable control group for CoCl2?
7. The tissue analysis photo is unclear. Please increase its resolution or use an
image analysis program for quantification.
8. Why study a Cobalt uptake in DPSC and BMSC cells? what does the unequal
uptake amount indicate?
9. All results need to be written as analytic figures. There is a need to statistically
show the difference between DPSC and BMSC and consider the meaning of the
difference.
10. In result 3, “The pellet morphology” is explained, but there is no figure.
11. The content that should be written in “Results” is entered in “Discussion”.
12. There is a need to explain more rationales for why “gene of Hox A5, Hox A7, and
Hox C10” was chosen among the HOX family.
13. In your conclusion, I wish there is more consideration the relationship between
Hox gene expression and plasticity.
Other comments
The authors have collected a lot of data but there is a major issue
1. with the presentation of the data.
2. Language
The introduction is poorly written, the background of the study should be clearly scripted in the introduction. Rewrite
the Materials section is also poorly presented, need to elaborate and expand the writing
The results section has so many iages and graphs, yet nothing is discussed, please discuss each image and graph clearly
the discussion is okay, try to related more and talk of the mechanistic perspective.
Conclusions rewrite
The novelty of this study and what makes it unique from the other studies should be clearly explained.
Comments on the Quality of English Language
Need major language editing
Author Response
“Plasticity Comparison of Two Stem Cell Sources with Different 2 Hox Gene Expression Profiles in Response to Cobalt Chloride 3 Treatment During Chondrogenic Differentiation.”
Comment1. The data presented in Fig. 5 and 6 regarding collagen Ⅱ and Ⅹ expression is important for understanding the differential response of DPSCs and BMSCs to hypoxic conditions. Including graphs that quantitatively depict these results can provide a visual representation of the differences in collagen expression level, making it easier to interpret the data.
Response 1. Based on valuable comments of reviewers we have added morphometrical analysis and graphs to collagen X and collagen II results.
Comment 2. The introduction provides a comprehensive overview of the challenges in cartilage regeneration and the potential of mesenchymal stem cells. However, it would be beneficial to provide more detailed information on the role of hypoxia in stem cell differentiation.
Response 2. According to the valuable comments of the reviewer, the role and major molecular mechanism of hypoxia by which exert the effects on chondrogenic differentiation have been elaborated in the introduction.
Comment 3. Your study highlights the differences in chondrogenic differentiation between DPSCs and BMSCs with different HOX gene expression profiles. In the introduction, it would be useful to include more detailed information on the role of HOX gene in stem cell, for example, in stem cell behavior, differentiation potential, plasticity, and responsiveness to environmental factors.
Response 3. More detailed explanations have been added to the introduction section to represent more detailed information on Hox genes and their roles in stem cells according to the valuable comments of reviewers.
Comment 4. The explanation of Hox gene within the Introduction part seems insufficient.
Response 4. To suffice, the descriptions on hox genes, the explanations on this subject have been expanded in the introduction.
Comment 5. In Fig. 1, what does the large size of BMSCs signify?
Response 5. The size of cells is different and is dependent on the source of isolation. Here the larger size of bone marrow-derived cells is obvious and has been mentioned just for the reader’s information. In our previous results we have seen different cell size isolated from adipose tissue, Wharton jelly and cord blood. Their tissue-dependent size is one of their characteristics and to best of our knowledge has no biological or molecular aspects during chondrogenic differentiation could be attributed to this variation. According to the reviewer’s question and as it did not have any significant effect on our experiments, the sentence about the size has been deleted.
Comment 6. Does CoCl2 play a role during pellet generation after pretreatment, and was there no comparable control group for CoCl2?
Response 6. In response to the reviewer’s question, it could be noted that all the experiments on CoCl2-treated samples have been accompanied by untreated samples which served as control group. The effect of CoCl2 on pellet formation has been elaborated under “pellet morphology” and “H&E staining” title in results section and it has been discussed in the discussion section, according to the reviewer’s comment.
Comment 7. The tissue analysis photo is unclear. Please increase its resolution or use an image analysis program for quantification.
Response 7. Based on valuable comments of reviewers we have added morphometrical analysis to collagen X and collagen II results. The original images have higher resolution, but merging multiple images in a single image decreases their quality. We have tried to increase their resolution as much as possible to meet the journal’s criteria.
Comment 8. Why study a Cobalt uptake in DPSC and BMSC cells? what does the unequal uptake amount indicate?
Response 8. In fact, ICP-MS was used in this study only to clarify whether superior chondrogenic response of DPSCs refers to higher amount of CoCl2 uptake or it could be attributed to their higher plasticity of DPSCs. As demonstrated in the ICP-MS results, DPSCs showed higher response to Cocl2 while their uptake for CoCl2 was less, compared to BMSCs. This could be interpreted that response to hypoxia mimicking agent is dependent on cell naïve properties, such as plasticity, rather than amount of CoCl2 in the cells. Based on the literature hypoxia mimicking-agents concentration could affect positively HIF-1a protein level in a concentration dependent manner, but this relation is controversial between hypoxia mimicking agents and chondrogenic differentiation. This is the reason for continuing research on this subject.
Comment 9. All results need to be written as analytic figures. There is a need to statistically show the difference between DPSC and BMSC and consider the meaning of the difference.
Response 9. Based on valuable comments of reviewers we have added morphometrical analysis to collagen X and collagen II results. In addition to MTT, ALP activity, ICP and qPCR, Alcian blue staining has been already quantified by GAG quantification. Figures related to cell and pellet morphology and integrity (figures 1, 3, 4) are gross qualitative images. P-values have been also added in the text of Results section for significant differences.
Comment 10. In result 3, “The pellet morphology” is explained, but there is no figure.
Response 10. Figure number 3 has been added according to the reviewer’s comment that exhibited pellet morphology.
Comment 11. The content that should be written in “Results” is entered in “Discussion”.
Response 11. Some sentences that were related to the results section have been removed from the discussion according to the reviewer’s comment.
Comment 12. There is a need to explain more rationales for why “gene of Hox A5, Hox A7, and Hox C10” was chosen among the HOX family.
Response 12. According to the reviewer’s comment It has been mentioned the reason for selecting these three genes at the end of introduction. It is noticeable that SYBR Green-Based Real-Time Quantitative PCR needed a pair of primers to meet specific criteria. These essential criteria restricted AllelID to design acceptable primers for all HOX genes, but Hox A5, Hox A7 and Hox C10.
Comment 13. In your conclusion, I wish there is more consideration the relationship between Hox gene expression and plasticity.
Response 13. Conclusion has been revised and elaborated on Hox genes to meet the valuable reviewer’s comments.
Other comments
The authors have collected a lot of data but there is a major issue
Comment 1. with the presentation of the data.
Response 1. Thanks to the valuable reviewer’s comments, it has been tried to enrich the paper by using quantitative analysis and representation of qualitative images. Some modifications have been also made in the statistical analysis and the results represented in new graphs.
Comment 2. Language
Response 2. The manuscript has been totally reviewed and edited for English language to meet higher standard of academic writing.
Comment 3. The introduction is poorly written, the background of the study should be clearly scripted in the introduction. Rewrite
Response 3. Based on the this and other comments which valuably recommended more detailed explanation and elaboration in the introduction, the text in this section has been rewritten and expanded according to the reviewer’s comments.
Comment 4. the Materials section is also poorly presented, need to elaborate and expand the writing
Response 4. Thanks to the comment of the reviewer, the material and methods section has been elaborated to make the procedures clearer with more details.
Comment 5. The results section has so many iages and graphs, yet nothing is discussed, please discuss each image and graph clearly
Response 5. According to the comments on the discussion, this section has been extended and more detailed explanations of the results have been added. Moreover, quantitative analysis has been added to the qualitative images to enrich the results.
Comment 6. The discussion is okay, try to related more and talk of the mechanistic perspective.
Response 6. The mechanistic perspective behind the obtained results have been discussed more according to the valuable reviewer’s comment, including hypoxia molecular pathway and Hox genes which could be the reasons behind our results.
Comment 7. Conclusions rewrite
The novelty of this study and what makes it unique from the other studies should be clearly explained.
Response 7. Conclusion has been revised and elaborated the novelty of the study to meet the valuable reviewer’s comments.
Comment 8. Comments on the Quality of English Language
Need major language editing
Response 8. The manuscript has been totally reviewed and edited for English language to meet higher standard of academic writing.
Reviewer 2 Report
Comments and Suggestions for Authors
In this manuscript, Khajeh et al. compared the differences in chondrogenic differentiation between two MSCs. The topic is interesting and might have significant translational potential. However, the current manuscript is superficially descriptive. In addition, it is not clear whether the differences between DPSCs and BMSCs hold if more cell lines are analyzed and more culture/differentiation conditions are applied. Below are my other comments.
1. More detailed information are expected for the DPSCs and BMSCs, such as the species, age, gender, etc. Would differences in these aspects contribute to the differences in chondrocyte differentiation? More biological replicates (more cell lines) should be tested. Also, for the other reagents used in this study, more information is expected, such as the catalog number or RRID for the antibodies.
2. Instead of using hypoxia mimicking conditions, are these results reproducible when the cells are cultured in 5% O2 hypoxia conditions?
3. Lines 225-227, please remove the instructions.
4. Figure 1, “Each arrow shows a stem cell.” How these cells with arrow are different from the remaining cells? How these “stem cell” characterized?
5. For all the IHC staining, please provide quantification results.
6. Figure 7, since the DPSCs and BMSCs show differential CoCl2 uptake amount, is it possible that the differences the authors observed might come from different hypoxia levels in these two cell types?
7. Figure 8, comparing DPSC-untreated and BMSC-treated is biologically meaningless, so are some comparisons in Figure 9.
8. I am assuming that DPSCs and BMSCs have many other differentially expressing genes, in addition to the three Hox genes the authors analyzed. Why special attention was paid to these three genes? Will overexpression of them in DPSCs or deletion of them in BMSCs revert their differentiation properties?
9. Line 410, “However, 100 µM for 24 hours was chosen for pretreatment of DPSCs and BMSCs to provide the same conditions during comparisons.” This claim is not supported by the data in Figure 7, where the authors showed that the CoCl2 concentrations in the two cell types are drastic different.
10. The text needs to be carefully proofread. The discussion section is hard to follow due to its lengthy and lack of logic. Please revise.
Comments on the Quality of English LanguageIn this manuscript, Khajeh et al. compared the differences in chondrogenic differentiation between two MSCs. The topic is interesting and might have significant translational potential. However, the current manuscript is superficially descriptive. In addition, it is not clear whether the differences between DPSCs and BMSCs hold if more cell lines are analyzed and more culture/differentiation conditions are applied. Below are my other comments.
1. More detailed information are expected for the DPSCs and BMSCs, such as the species, age, gender, etc. Would differences in these aspects contribute to the differences in chondrocyte differentiation? More biological replicates (more cell lines) should be tested. Also, for the other reagents used in this study, more information is expected, such as the catalog number or RRID for the antibodies.
2. Instead of using hypoxia mimicking conditions, are these results reproducible when the cells are cultured in 5% O2 hypoxia conditions?
3. Lines 225-227, please remove the instructions.
4. Figure 1, “Each arrow shows a stem cell.” How these cells with arrow are different from the remaining cells? How these “stem cell” characterized?
5. For all the IHC staining, please provide quantification results.
6. Figure 7, since the DPSCs and BMSCs show differential CoCl2 uptake amount, is it possible that the differences the authors observed might come from different hypoxia levels in these two cell types?
7. Figure 8, comparing DPSC-untreated and BMSC-treated is biologically meaningless, so are some comparisons in Figure 9.
8. I am assuming that DPSCs and BMSCs have many other differentially expressing genes, in addition to the three Hox genes the authors analyzed. Why special attention was paid to these three genes? Will overexpression of them in DPSCs or deletion of them in BMSCs revert their differentiation properties?
9. Line 410, “However, 100 µM for 24 hours was chosen for pretreatment of DPSCs and BMSCs to provide the same conditions during comparisons.” This claim is not supported by the data in Figure 7, where the authors showed that the CoCl2 concentrations in the two cell types are drastic different.
10. The text needs to be carefully proofread. The discussion section is hard to follow due to its lengthy and lack of logic. Please revise.
Author Response
Comment. In this manuscript, Khajeh et al. compared the differences in chondrogenic differentiation between two MSCs. The topic is interesting and might have significant translational potential. However, the current manuscript is superficially descriptive. In addition, it is not clear whether the differences between DPSCs and BMSCs hold if more cell lines are analyzed and more culture/differentiation conditions are applied. Below are my other comments.
Response. According to the reviewer’s comments morphometric quantitative analysis has been added to descriptive immunohistochemical results to provide the possibility of quantitative representation of data. In addition, statistical analysis has been modified for some of the results, according to the reviewer’s comment, which led to more precise results and different representation of graphs. It is notable that each experiment was performed at least on three independent biological replications.
We have mentioned it in the material and methods section, according to the reviewer’s comment as: All data resulted from at least three independent experiments.
Comment 1. More detailed information are expected for the DPSCs and BMSCs, such as the species, age, gender, etc. Would differences in these aspects contribute to the differences in chondrocyte differentiation? More biological replicates (more cell lines) should be tested. Also, for the other reagents used in this study, more information is expected, such as the catalog number or RRID for the antibodies.
Response 1. These cells have been dedicated by two expert research centers with intensive experiences in MSCs isolation and characterization. These cells have been previously isolated from human and characterized by osteogenic and adipogenic differentiation and flowcytometry for CD markers by these research centers. The human source of the cells and age of donors has been added to the materials and methods section. All the tests have been performed on at least three independent biological replicates for DPSCs and for BMSCs to reduce the possible effects of variation between samples (cells) on the results.
More information for utilized reagents in this study has been added to the Materials and Methods section.
Comment 2. Instead of using hypoxia mimicking conditions, are these results reproducible when the cells are cultured in 5% O2 hypoxia conditions?
Response 2. In this study one of the aims was to provide hypoxic conditions through hypoxia mimicking agent that presents a simple in use and cost-effective method compared to applying low oxygen tension which needs specialized incubators and chambers. Corresponding of CoCl2 concentration in this method to the level of oxygen in low oxygen tension condition was not possible for us and could be the aim of another study in the future to answer this valuable question. Moreover, hypoxia mimicking agent provision of hypoxia level stability, based on the literature.
Comment 3. Lines 225-227, please remove the instructions.
Response 3. Thanks to the concise comment of the reviewer this instruction has been removed.
Comment 4. Figure 1, “Each arrow shows a stem cell.” How these cells with arrow are different from the remaining cells? How these “stem cell” characterized?
Response 4. Thanks to the reviewer’s comment. The arrows were used just to show representative stem cells in each image among the other stem cells and there is no noticeable morphologic difference between individual stem cells in each image. These cells have been dedicated by two expert research centers with intensive experiences in MSCs isolation and characterization. These cells have been previously characterized by osteogenic and adipogenic differentiation and flowcytometry for CD markers by these two centers.
Comment 5. For all the IHC staining, please provide quantification results.
Response 5. Based on valuable comments of reviewers we have added morphometric analysis to the qualitative collagen X and collagen II results to provide the possibility of quantitative comparisons.
Comment 6. Figure 7, since the DPSCs and BMSCs show differential CoCl2 uptake amount, is it possible that the differences the authors observed might come from different hypoxia levels in these two cell types?
Response 6. In fact, ICP-MS was used in this study only to clarify whether superior chondrogenic response of DPSCs refers to higher amount of CoCl2 uptake or it could be attributed to their higher plasticity of DPSCs. As demonstrated in the ICP-MS results, DPSCs showed higher response to Cocl2 while their uptake for CoCl2 was less compared to BMSCs. This could be interpreted that response to hypoxia mimicking agent is dependent on cell naïve properties, such as plasticity, rather than amount of CoCl2 in the cells. Based on the literature hypoxia mimicking-agents concentration could affect positively HIF-1a protein level in a concentration dependent manner, but this relation is controversial between hypoxia mimicking agents and chondrogenic differentiation. This is the reason for continuing research in this field.
Comment 7. Figure 8, comparing DPSC-untreated and BMSC-treated is biologically meaningless, so are some comparisons in Figure 9.
Response 7. According to valuable comments of the reviewers the analysis for ALP activity and GAG production have been modified and the comparison between DPSC-untreated and BMSC-treated has been removed from the results and accordingly the related texts in the paper have been modified.
Comment 8. I am assuming that DPSCs and BMSCs have many other differentially expressing genes, in addition to the three Hox genes the authors analyzed. Why special attention was paid to these three genes? Will overexpression of them in DPSCs or deletion of them in BMSCs revert their differentiation properties?
Response 8. According to the reviewer’s comment It has been mentioned the reason for selecting these three genes at the end of introduction. It is noticeable that SYBR Green-Based Real-Time Quantitative PCR needed a pair of primers to meet specific criteria. These essential criteria restricted AllelID to design acceptable primers for all Hox genes, but Hox A5, Hox A7 and Hox C10.
The reviewer’s suggestion to further overexpressing of these genes in stem cells is noticeable and could shed more lights on their individual roles in the chondrogenic differentiation, but it was not in the scope of our study, which explore primarily whether there is differences between these Hox positive and Hox negative cell source in chondrogenic differentiation efficiency.
Comment 9. Line 410, “However, 100 µM for 24 hours was chosen for pretreatment of DPSCs and BMSCs to provide the same conditions during comparisons.” This claim is not supported by the data in Figure 7, where the authors showed that the CoCl2 concentrations in the two cell types are drastic different.
Response 9. It is an approved and widely used approach to administer the same concentration of chemical/pharmaceutical agents in the cell culture for comparison their effects on different groups, what has been also performed in the same manner in our study.
In fact, ICP-MS was used in this study only to clarify whether superior chondrogenic response of DPSCs refers to higher amount of CoCl2 uptake or it could be attributed to their higher plasticity of DPSCs. As demonstrated in the ICP-MS results, DPSCs showed higher response to Cocl2 while their uptake for CoCl2 was less compared to BMSCs. This could be interpreted that response to hypoxia mimicking agent is dependent on cell naïve properties, such as plasticity, rather than amount of CoCl2 in the cells.
Comment 10. The text needs to be carefully proofread. The discussion section is hard to follow due to its lengthy and lack of logic. Please revise.
Response 10. Thanks to the reviewer’s comment, the discussion has been rewritten, some texts have been omitted and some explanations, as reviewers recommended, have been added to the discussion. In addition, the discussion has been reorganized according to the results’ arrangements.
Reviewer 3 Report
Comments and Suggestions for Authors
The presented paper is a study on the potential of cell stimulation for articular cartilage regeneration, highlighting the limitations of self-repair in cartilage and the need for effective regenerative therapies. The study concludes that the unique molecular characteristics of DPSCs, particularly their Hox-negative expression pattern, endow them with superior chondrogenic differentiation potential and plasticity. This could make DPSCs a promising candidate for future regenerative minimally-invasive therapies aimed at cartilage regeneration.
Some important comments:
1. Line 93: The authors used the term "mesenchymal stem cells", but the study seems to use the cells called "mesenchymal stromal cells". I know that earlier studies did not distinguish between mesenchymal stem cells and mesenchymal stromal cells (bone marrow-derived (BM-MSCs), but current approaches do not allow to call these multipotent cells "stem cells" [https://pubmed.ncbi.nlm.nih.gov/20737049/]. I recommend that the authors reconsider the terminology used and refer to the cells used as 'mesenchymal stromal cells'.
2. Line 94: "DPSCs were obtained from the Stem Cells Technology Research Center, while BMSCs were acquired from the Bonyakhteh Research Center" - the cell source (source number or biobank ID or written patient consent) and phenotype should be disclosed in detail. Were they both human cells?
3. Line 118: "2.5×105 MSC" - degree of number.
4. Line 295: Figures 5 and 6: "In general, the expression of Collagen II in samples differentiated from BMSCs was higher in both normoxic and hypoxic induction compared to DPSC samples." - Quantitative morphometric methods are needed to prove states like this.
5. Statistical analysis: Figures 8 and 9: One-way ANOVA is not applicable to analyze these data, in this case two-way ANOVA is required.
6. Line 353: It appears that cobalt chloride (CoCl2) is a hypoxia mimetic agent that could induce cellular senescence in MSCs [https://pubmed.ncbi.nlm.nih.gov/37047346/]. I recommend disclosing this issue in the Discussion section.Author Response
The presented paper is a study on the potential of cell stimulation for articular cartilage regeneration, highlighting the limitations of self-repair in cartilage and the need for effective regenerative therapies. The study concludes that the unique molecular characteristics of DPSCs, particularly their Hox-negative expression pattern, endow them with superior chondrogenic differentiation potential and plasticity. This could make DPSCs a promising candidate for future regenerative minimally-invasive therapies aimed at cartilage regeneration.
Some important comments:
Comment 1. Line 93: The authors used the term "mesenchymal stem cells", but the study seems to use the cells called "mesenchymal stromal cells". I know that earlier studies did not distinguish between mesenchymal stem cells and mesenchymal stromal cells (bone marrow-derived (BM-MSCs), but current approaches do not allow to call these multipotent cells "stem cells" [https://pubmed.ncbi.nlm.nih.gov/20737049/]. I recommend that the authors reconsider the terminology used and refer to the cells used as 'mesenchymal stromal cells'.
Response 1. Thanks to the precise comment of the reviewer and the exploring some papers until 2024 (such as https://doi.org/10.1038/s41392-023-01338-2) it has been found that the bone mesenchymal stem cell term should be used more cautiously and express as bone mesenchymal stem/stromal cells. It has been corrected in the text according to the reviewer’s comment.
Comment 2. Line 94: "DPSCs were obtained from the Stem Cells Technology Research Center, while BMSCs were acquired from the Bonyakhteh Research Center" - the cell source (source number or biobank ID or written patient consent) and phenotype should be disclosed in detail. Were they both human cells?
Response 2. These cells have been dedicated by two expert research centers with intensive experiences in MSCs isolation and characterization. These cells have been previously isolated from human subjects and characterized by osteogenic and adipogenic differentiation and flowcytometry for CD markers by these research centers.
Comment 3. Line 118: "2.5×105 MSC" - degree of number.
Response 3. It has been corrected according to the reviewer’s comment.
Comment 4. Line 295: Figures 5 and 6: "In general, the expression of Collagen II in samples differentiated from BMSCs was higher in both normoxic and hypoxic induction compared to DPSC samples." - Quantitative morphometric methods are needed to prove states like this.
Response 4. Thanks to valuable comments of reviewers, we have added morphometrical analysis to collagen X and collagen II results to meet the need for quantitative comparison. The text for Immunohistochemistry of collagen II and X has been corrected and the morphometry results for collagen II and X statistically present in the morphometry section.
Comment 5. Statistical analysis: Figures 8 and 9: One-way ANOVA is not applicable to analyze these data, in this case two-way ANOVA is required.
Response 5. Thanks to this the valuable reviewer’s comment, the analysis for ALP activity, GAG quantification and newly added morphometric analysis of collagen II and collagen X have been performed by two-way ANOVA and accordingly the related texts in the paper have been modified.
Comment 6. Line 353: It appears that cobalt chloride (CoCl2) is a hypoxia mimetic agent that could induce cellular senescence in MSCs [https://pubmed.ncbi.nlm.nih.gov/37047346/]. I recommend disclosing this issue in the Discussion section.
Response 6. This paper has been cited and mentioned in the discussion section of manuscript, as recommended by the reviewer.
Round 2
Reviewer 1 Report
Comments and Suggestions for Authors
Can be accepted
Author Response
Comment 1: Can be accepted
Response 1: I would like to thank the respectable reviewer for the scientific comments which certainly pave the way towards the publication of this manuscript.
Reviewer 2 Report
Comments and Suggestions for Authors
The authors have addressed my comments. The manscript is improved and I appreciate their effort.
Author Response
Comment 1: The authors have addressed my comments. The manuscript is improved and I appreciate their effort.
Response 1: I would like to thank the respectable reviewer for the scientific comments which certainly pave the way towards the publication of this manuscript.
Reviewer 3 Report
Comments and Suggestions for Authors
The authors responded satisfactorily to all my comments and made all the necessary changes to the manuscript. However, some questions occurred to me while reading the revised manuscript:
1. Simple summary has been improved, but looks weak: The first sentence looks trivial and should be rewritten ("It is difficult for body to heal damaged joint cartilage injuries.")
2. Line 1349: "Application of MSCs may potentially make cartilage repair more widely available" - It should be added that the authors are talking about autologous cells.
Author Response
Comment: The authors responded satisfactorily to all my comments and made all the necessary changes to the manuscript. However, some questions occurred to me while reading the revised manuscript:
Response: I would like to thank the respectable reviewer for the scientific comments which certainly pave the way towards the publication of this manuscript.
Comment 1. Simple summary has been improved, but looks weak: The first sentence looks trivial and should be rewritten ("It is difficult for body to heal damaged joint cartilage injuries.")
Response 1. According to the valuable comment of the reviewer, the simple summary has been totally revised (highlighted) and the mentioned sentence by the reviewer has been corrected in the summary text.
Comment 2. Line 1349: "Application of MSCs may potentially make cartilage repair more widely available" - It should be added that the authors are talking about autologous cells.
Response 2. According to the reviewer's valuable comment it has been done and highlighted in the text.